# Adaptive Doppler Compensation for Mitigating Range Dependence in Forward-Looking Airborne Radar

**Muhammad Bilal Khan [1,†], Ahmed Hussain [2,†], Umar Anjum [2], Channa Babar Ali [2] and Xiaodong Yang [1,\*]**

[1]  School of Electronic Engineering, Xidian University, Xi'an 710071, China; bilal@stu.xidian.edu.cn
[2]  Aerospace & Aviation Campus Kamra, Air University, Attock 43570, Pakistan;
    185128@students.au.edu.pk (A.H.); 185127@students.au.edu.pk (U.A.); 185125@students.au.edu.pk (C.B.A.)
[\*]  Correspondence: xdyang@xidian.edu.cn
[†]  These authors contributed equally to this work.

**Abstract:** In this paper, we present ground moving target indication (GMTI) signal processing algorithm encompassing clutter suppression, target detection and parameter estimation. One of the most significant yet least publicized is the need of the GMTI mode for a forward-looking airborne radar. The integration of GMTI mode in a forward-looking airborne radar allows reconnaissance and surveillance operations in all weather conditions. In this context, space time adaptive processing (STAP) offers a unique prospect of enabling the GMTI mode in forward looking airborne radar. STAP is a two-dimensional filter designed to suppress platform motion-induced clutter Doppler spread. Interference is characterized by a covariance matrix. In the case of a forward-looking airborne radar, the clutter Doppler is dependent on range. Clutter Doppler dependency on the range renders the training cells heterogeneous. The heterogeneity effects are particularly prominent in the near range bins. Non-homogeneous training cells have a deleterious effect on STAP performance. In this study, we propose an adaptive Doppler compensation to mitigate the degraded STAP performance in the near range bins. The adaptivity feature circumvents the need for the availability of radar parameters in real-time. The real time implementation of STAP is impeded by requirements of a large number of training samples and covariance matrix inversion. Therefore, there is a dire need to devise a framework to detect and estimate target parameters within the STAP. In this regard, we propose an efficient STAP algorithm to detect and estimate target parameters. STAP weights are applied to the input data to obtain a 3D array. The range projection of the 3D array is utilized to detect and estimate the range of the target, while the angle–Doppler projection is used to estimate spatial and temporal parameters of the target. Most of the literature on STAP is geared towards a known covariance matrix. The assumption of a known covariance matrix may degrade STAP performance because of the inherent mismatches between the actual and assumed target steering vectors. In this study, we estimate the covariance matrix based on the synthetic data generated from a model of an airborne phased array radar. The developed STAP algorithms closely mimic a real-time implementation scheme in an airborne radar platform. The results of the proposed algorithm are validated through target parameter estimation and STAP metrics on synthetic data.

**Keywords:** space-time adaptive processing; adaptive Doppler compensation; range dependence

## 1. Introduction

Recent advancements in airborne phased array radars have enabled agile beam steering and high-speed multimode processing. The groundbreaking revolutions in this field have established a new

paradigm in battlefield management and strategic bombing missions. The bombing operations involve detection, tracking and identification of enemy vehicles, tanks and convoys. Presently, such operations are conducted with thermal or laser designated weapons. However, the performance of these systems deteriorates in cloudy weather and also requires a direct line of sight. The GMTI radar mode can discriminate slow-moving ground targets from heavy clutter at shallow slant angles and in all weather conditions [1]. Furthermore, the GMTI radar allows a larger footprint with minimal loss in accuracy. This mode of radar could also be utilized for civil applications such as surveillance during emergency evacuations, border management and traffic management [2].

One of the simplest approaches to detect a moving target is to incorporate a high pass filter after pulse compression [3]. This is a non-adaptive approach in which slow-moving targets are attenuated that appears close to the background clutter. In displaced phase center antenna (DPCA), platform velocity and pulse repetition frequency (PRF) are adjusted so that clutter appears stationary that can be attenuated using two-pulse canceller architecture. The DPCA technique restricts platform velocity and PRF, which may not be a viable option in real-time radar operations [4]. Along track interferometry (ATI) entails utilization of radar images from various look angles to decipher minute phase variations [5,6]. The manipulation of radar images at different time instants can indicate the presence of slow-moving targets. Space time adaptive processing (STAP) possesses tremendous potential in such a scenario and can be implemented in airborne phased array radar.

Clutter rejection by the STAP filter is motivated by the clutter correlation in spatial and temporal dimensions [7–9]. STAP refers to two-dimensional filtering of clutter and jamming interference. The spatial dimension is afforded by the antenna elements while temporal dimensional is exploited by transmitting multiple pulses in a coherent processing interval (CPI). A space–time processor may be conceived as a combined Doppler filter and beamformer. This combined filter places nulls in the direction of a jammer and interference while maximizing gain in the assumed direction and Doppler of the target.

Interference in a particular range bin is characterized by averaging the covariance matrices from the surrounding range bins [10]. The accuracy of the covariance matrix for the particular range bin is strongly reliant on the availability of homogeneous training cells. Clutter Doppler in case of a side-looking radar is stationed at zero frequency and has desirable characteristics with regards to adaptive processing [11].

In the case of a forward-looking airborne radar, clutter Doppler is dependent on the range and each range bin is associated with a different Doppler frequency [12–14]. The heterogeneous training range bin samples yield an erroneous covariance matrix estimate, which may degrade STAP performance in terms of the increased minimum detectable velocity (MDV) and decreased usable Doppler spread fraction (UDSF).

Isodops are a set of points with the same Doppler frequency. Likewise, contours of the constant range are termed as iso-range rings. A set of hyperbolas depicting isodops has been illustrated in Figure 1a while iso-range rings overlaid on the isodops are shown in Figure 1b. It can be observed that Doppler frequency remains constant for cross flight direction while it changes with range for any other geometry. It is pertinent to note that the problem of range dependence exists in all geometries except a perfectly side-looking one. The rate of change of Doppler is different for each of the radar geometry. However, the Doppler dependence on range is highest in the case of a forward-looking radar Moreover, the rate of change of Doppler is greater at near ranges and it tends to lessen as the range increases.

After target detection, parameter estimation is one of the critical tasks of airborne radar. Traditionally, Doppler processing is employed to estimate target velocity and the bearing of the detected target is coarsely defined by the antenna pointing direction. Monopulse feed network is used in airborne intercept radars to further improve the angle estimate. However, the performance of monopulse degrades due to distortions introduced by adaptive beamforming [15,16]. Although, STAP filter aids in slow-moving target detection, yet it offers another perspective to estimate target

parameters. A STAP radar can estimate the target parameters by finding the maximum likelihood estimate in the angle–Doppler domain [17]. It may be known that STAP per se is computationally expensive hence an efficient framework is required to undertake target detection and parameter estimation efficiently.

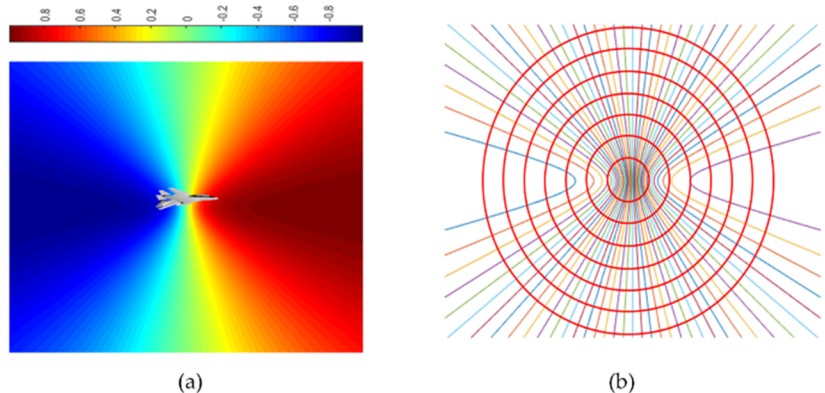

**Figure 1.** (**a**) Clutter isodops and (**b**) range rings overlaid on clutter isodops.

### 1.1. Related Work

The issue of Doppler dependence on the range in the case of forward-looking radars has been investigated by many researchers. The degraded performance due to the heterogeneity of training samples can be alleviated by reduced dimension (RD) STAP algorithms [18–20]. Organization of RD STAP algorithms is depicted in Figure 2. RD STAP algorithms can be categorized into data-independent or data-dependent transformation. The data dependence or independence relates to whether preprocessing takes into account the covariance matrix estimate. Data independent transformations include pre-Doppler and post-Doppler algorithms. Data dependent transformation is further classified into signal dependent and signal independent. In the signal independent transformation, Eigen analysis of the covariance matrix excluding the target steering vector is undertaken. One of the popular examples of a signal independent rank reducing technique is the principle component analysis (PCA) [21]. It is highlighted that PCA based methods have failed to garner application in practical systems due to high computational cost. Furthermore, performance of this technique is exacerbated due to decorrelation effects such as range walk and internal clutter motion. Decorrelation effects increase the number of degrees of freedom spanned by clutter subspace. Data dependent and signal dependent methods include the effect of target steering vector in adaptive data transformation. Multistage Weiner filter is an example of signal dependent transformation [22]. Weights in multistage Weiner filter are formed by an inner product of the inverse of the cross correlation matrix and target steering vector. Presented another way, an optimization problem is solved in which the mean square error for difference of weight vector and target steering vector is minimized in each stage.

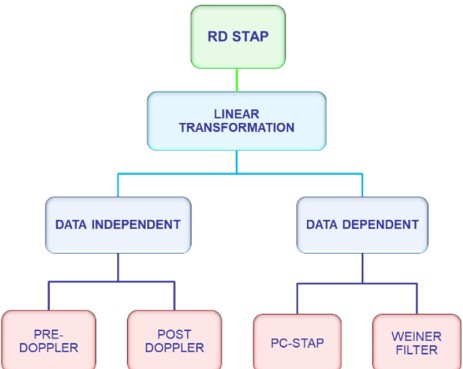

**Figure 2.** Reduced dimension space time adaptive processing (STAP) algorithms.

RD STAP algorithms reduce the requisite number of training samples and thereby attempt to minimize angle–Doppler variation. However, an optimal solution may not be achieved, since the root cause of the problem is not addressed directly.

In the time-varying weights method, the weight vector is assumed to be a linear function of fast time [23,24]. The required number of training samples is increased, which may prove computationally expensive. Borsari et al. [25] first proposed a heuristic approach in which the Doppler frequency of the training cells is shifted based on the Doppler frequency of the reference cell. In this method, it is assumed that radar parameters that determine the Doppler frequency for each cell are known a priori. However, radar parameters like velocity and platform height provided by other aircraft sensors may be corrupted due to noisy measurements. It may be noted that the Doppler warping shifts Doppler for a single look angle. Doppler compensation for multiple look angles is undertaken in the higher order Doppler warping (HODW) method [26]. In the registration-based compensation method, the clutter angle Doppler spectrum for each range bin is estimated and then curve fitting is used to align the clutter power spectrum of the training cell with that of the cell under test [27]. In case of bistatic radar geometry, clutter spectrum alignment in both Doppler and spatial dimension is required to be undertaken. Angle Doppler compensation (ADC) aligns the clutter power spectrum in Doppler and the spatial domain for a single look angle [28]. Similarly, in the derivative based update (DBU) method, Taylor series expansion of weight vector is used in which the constant and linear terms are retained only [29]. It is assumed that weights are the linear function of the range. DBU consumes more computational power due to the requirement of higher number of weight vectors.

Target parameter estimation along with detection in a STAP radar has been a topic of interest for many researchers. Reed et al. [30] presented a joint constant false alarm rate (CFAR) detection and target estimation framework. It was proposed that target detection might be carried out in the angle–Doppler domain. Montanari et al. [31] reported that the maximum likelihood angle estimator has superior performance compared to a generalized monopulse in the case of STAP. Zhou et al. [32] extended the generalized monopulse to STAP in order to minimize the beam distortions and hence better angle estimation.

### 1.2. Main Contribution of This Paper

Most of the published results on STAP are founded on the premise of a known covariance matrix for a given range bin. However, in reality, the covariance matrix is unknown a priori and requires to be estimated through surrounding training cells. Furthermore, target Doppler and angle of arrival are also assumed to be known. Mismatches in the assumed and actual target parameters degrade STAP performance. Furthermore, this assumption does not allow estimation of the target Doppler and angle. In this paper, an airborne X-band phased array radar has been modeled to generate synthetic data. Moreover, the covariance matrix is estimated from the generated data instead of employing a theoretical model of the covariance matrix.

Furthermore, a joint STAP framework is presented in which target detection and parameter estimation is carried out in an efficient way. We propose an adaptive Doppler compensation method to mitigate clutter Doppler dependence on the range for a forward-looking airborne radar. The proposed method circumvents the need for prior information about radar parameters. Spectrum alignment of training cells is carried out by adaptively compensating the Doppler frequency of each range bin. Although, results of proposed adaptive Doppler compensation for a forward-looking geometry are presented here, however the methodology is also applicable for all the non-side looking geometries.

The process of application of STAP weights on the input data results in the suppression of clutter. However, a detection algorithm is required so as to declare the presence or absence of a target. In this regard, we propose non-coherent integration to convert STAP output data into fast time data. Then, the fast time data is subjected to 1D CFAR to report the range if the target is declared present. The target angle and velocity information is accrued by isolating peaks in the angle–Doppler domain of the STAP processed data.

*1.3. Organization of This Paper*

The airborne radar signal environment and angle–Doppler response for a forward-looking airborne radar are explained in Section 2. Besides, clutter Doppler dependency on the range has been expressed mathematically. Section 3 presents a brief overview of the optimum STAP processor. The adaptive Doppler compensation algorithm is proposed in Section 4. In Section 5, simulation results for the proposed algorithm have been produced and analyzed. Section 6 culminates with concluding remarks.

## 2. Angle Doppler Response

We considered a forward-looking airborne radar with a uniform linear array (ULA), to develop the clutter model and STAP algorithms. It is assumed that the radar platform moves with a constant velocity $v$ along the $x$-axis and maintains a constant height H. A generic airborne radar orientation depicting the geometrical parameters that affect the clutter spectrum is shown in Figure 3. The radar antenna illuminates a stationary scatterer $S$ on the ground, which is located at a slant range R. The scatterer subtends azimuth angle $\Phi$ and elevation angle $\theta$ with the radar beam. The scatterer is localized on the ground with respect to the radar beam and velocity $v$ in terms of angle $\beta$ and $\alpha$ respectively. Antenna array axis is defined by the dotted line with the bigger dots representing the individual antenna elements. The orientation of the antenna array with respect to an aircraft velocity vector is characterized by crab angle $\gamma$. The look angle $\beta$ varies as the phased array radar steers the antenna beam while the crab angle $\gamma$ remains constant for any particular airborne radar platform.

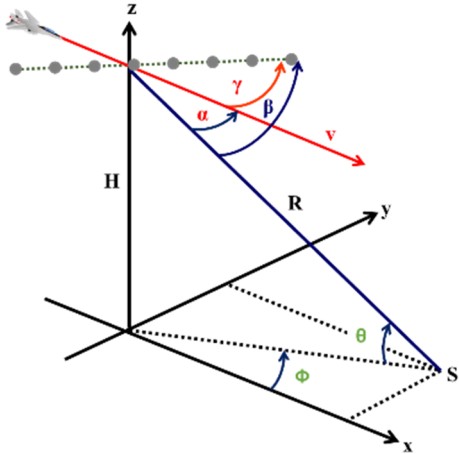

**Figure 3.** Geometry of the forward-looking airborne radar.

The fundamental relation that governs the Doppler frequency $f_d$ of a scatterer is given by the following equation:

$$f_d = \frac{2v}{\lambda} \cos(\Phi) \cos(\theta), \tag{1}$$

$$f_r = \cos(\Phi) \cos(\theta) \tag{2}$$

Normalized Doppler frequency $f_r$ is obtained by normalizing (1) with maximum Doppler $\frac{2v}{\lambda}$. Cosine of cone angle $\alpha$ is equal to the product of cosine of azimuth angle $\Phi$ and elevation angle $\theta$. Doppler dependency of stationary scatterer on the look angle $\beta$ and the crab angle $\gamma$ is achieved after some mathematical manipulations [33].

$$f_r^2 - 2f_r \cos(\gamma) \cos(\beta) + \cos^2 \beta - \sin^2 \gamma \cos^2 \theta = 0 \tag{3}$$

The crab angle differentiates the geometry of a front-looking radar from a side-looking radar. Crab angle is $90^\circ$ for a front-looking radar while it is $0^\circ$ for a side-looking radar. Incorporating the crab

angle values in (3) for a purely side-looking and front-looking radar geometry yields much simpler and intuitive equations.

$$f_r = \cos(\beta), \tag{4}$$

$$f_r = \sqrt{\left(1 - \left(\frac{R}{H}\right)^2\right) - (\cos\beta)^2} \tag{5}$$

Clutter Doppler frequency in (4) is predicated by a look angle for a side-looking geometry while range also affects the Doppler frequency for a forward-looking geometry as illustrated by (5). Clutter ridge appears as a diagonal line in the angle–Doppler domain for a side-looking radar. On the contrary, the linear relationship between Doppler and angle does not hold in the forward-looking radar. Clutter ridge manifests itself in the form of rotated ellipses for non-side looking radar geometries. The clutter spectrum for different range to height ratios (R/H) in the angle–Doppler domain for a forward-looking radar is depicted in Figure 4. It can be observed that the clutter spectra for different ranges do not align with each other. Assuming no range dependence, all the circles depicting clutter spectrum in Figure 4 should have same radius. However, due to clutter spectra variation, a family of circles spread over a swath was obtained, averaging of which will yield a dilated estimate of covariance matrix estimate for the cell under test.

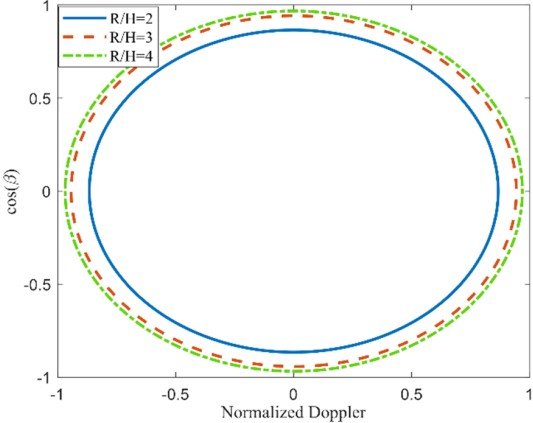

**Figure 4.** Clutter spectrum.

It is paramount to highlight that circles in Figure 4 are concentric. It can be inferred that spectrum alignment is required in Doppler dimension only for monostatic radar geometries. Clutter Doppler frequency obtained for different ranges, normalized to platform height for a forward-looking radar is depicted in Figure 5.

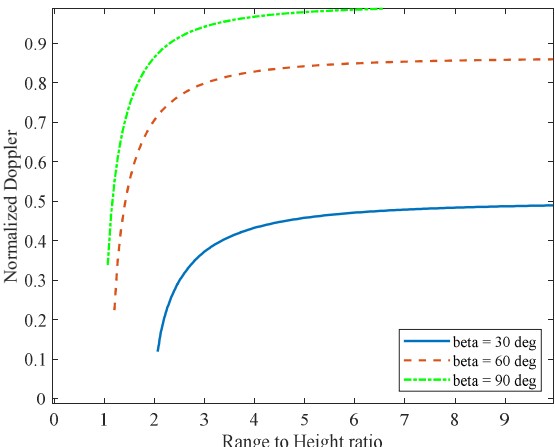

**Figure 5.** Clutter Doppler frequency at different ranges.

Doppler gradient is higher at near ranges and it settles to a constant value at higher ranges. Moreover, the Doppler gradient at near ranges is maximum when the radar antenna directs the beam in boresight direction while it decreases as the beam is steered away from the boresight.

### 3. Optimum Processor

The IQ samples received by a phased array radar can be visualized in the form of a data cube. The three dimensions of the radar data cube include fast time, slow time and a spatial axis. An airborne radar transmits a series of pulses towards the target and waits for the received echoes. The fast time axis represents the received samples of a single pulse from all ranges while the slow time axis depicts the same range sample received from different pulses. Since, a phased array radar has multiple receive channels, the spatial axis indicates the data samples with spatial phase shifts received by the individual channel.

Radar data cube comprises $M$ pulses and $N$ antenna elements in a single coherent processing interval (CPI). The data samples in the CPI are concatenated in the form of $NM \times 1$ space–time snapshot. The total number of range bins is determined by the product of sampling rate $Fs$ and pulse repetition interval (PRI). Interference covariance matrix $\boldsymbol{R_c}$ is formed by the inner product of the space–time snapshot $\boldsymbol{x}$ with its conjugate transpose as given by (6). The total number of training bins $L$ is determined by the Reed-Mallett-Brennan (RMB) rule for optimum performance [34].

$$\boldsymbol{R_c} = \frac{1}{L} \sum_{i=1}^{L} x_i x_i^H \tag{6}$$

The desired target response is formed by Doppler and spatial steering vectors. Target spatial steering vector $\boldsymbol{a}(\Psi)$ and Doppler steering vector $\boldsymbol{b}(w)$ are given by

$$\boldsymbol{a}(\Psi) = e^{j(2\Pi \frac{d}{\lambda} \cos \varnothing \cos \theta)(0:N-1)}, \tag{7}$$

$$\boldsymbol{b}(w) = e^{j(2\Pi(\frac{2v}{\lambda})\frac{1}{\text{PRF}})(0:M-1)} \tag{8}$$

where $d$ is spacing between antenna elements and $\lambda$ is the operating wavelength. Desired target response $v$ is formulated by the Kronecker product of target Doppler and spatial steering vectors. The target response vector $v$ is analogous to a search vector with a particular velocity and direction. The target search is carried out in the antenna beam pointing direction and for all the probable Doppler frequencies in a STAP radar.

$$\boldsymbol{v}(\Psi, w) = \boldsymbol{b}(w) \bigotimes \boldsymbol{a}(\Psi) \tag{9}$$

The inner product of the covariance matrix inverse and the desired target response results in the weight vector $\boldsymbol{w}$. Dimensions of the weight vector are the same as the input data snapshot.

$$\boldsymbol{w} = k\boldsymbol{R_c}^{-1}\boldsymbol{v} \tag{10}$$

In (10), $k$ is the normalization constant. The estimation accuracy of the covariance matrix is critical to optimum STAP performance. In the covariance matrix estimation, it is assumed that interference in the training range bins reflects the interference present in the cell under test. However, if the clutter spectrum varies in adjacent range cells, the estimated covariance matrix may not depict the actual interference characteristics. The angle–Doppler response of the weight vector without mitigating spectrum disparity in the training range bins will encompass a larger clutter area. A large clutter area in the angle–Doppler domain entails a wider null, which may extenuate potential targets. STAP weight vector response without data compensation is depicted in Figure 6. A wider clutter ridge in the form of a semicircle embodies the problem in hand.

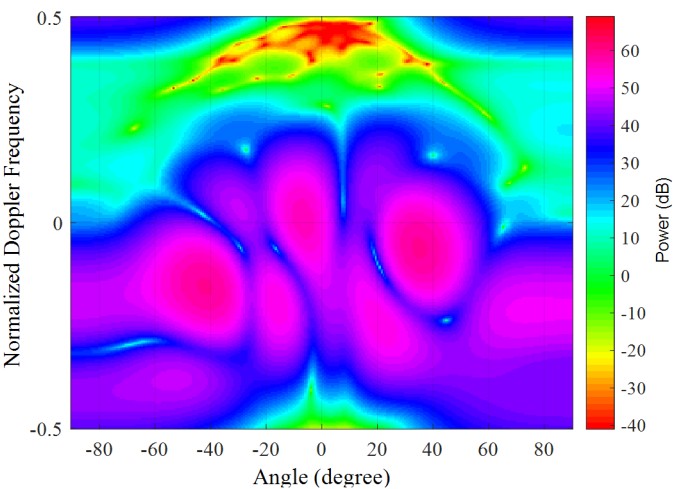

**Figure 6.** Weight vector response prior to Doppler compensation.

## 4. Adaptive Doppler Compensation

Clutter angle Doppler spectrum varies over the range for a forward-looking monostatic radar. In this case, the Doppler frequency of the clutter is a strong function of range, especially at near ranges. The proposed algorithm as illustrated in Figure 7 aims to align the clutter spectrum in the Doppler dimension. We proposed estimation of the Doppler frequency for each range bin. It is assumed that the training cells are free of any target. A non homogeneity test of the generalized inner product was used to detect and counter the presence of target like cells in the training region The Doppler estimation is undertaken by employing fast Fourier transform (FFT) along the slow time axis. Since there are multiple channels, the Doppler estimate for each channel is computed separately. Net Doppler estimate $f_{comp}$ for a given range bin is found by averaging Doppler frequencies for all channels. It is expected that the averaging process will reduce errors due to mismatches in the array channels.

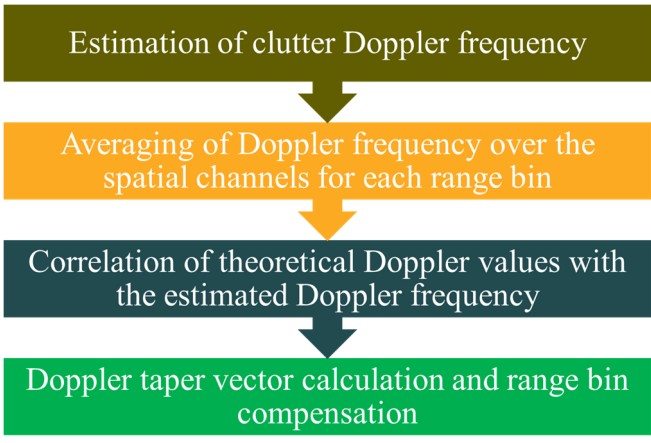

**Figure 7.** Proposed adaptive Doppler compensation method.

Estimated Doppler of each range cell will be correlated with the theoretical Doppler values given by Doppler warping. The purpose of this step is to ensure that estimated values of the clutter Doppler do not deviate too much from the theoretical values. In case of any deviation, the concerned training cell will be excluded from the training process, since deviation could be due to the presence of any target like artifact. Next a Doppler vector is formed in which Doppler frequency of each range bin is subtracted from the Doppler value of the cell under test.

A Doppler vector $d(w)$ for each range-bin is formed by:

$$d(w) = e^{j\left(2\Pi f_{comp}\left(\frac{2v}{\lambda}\right)\frac{1}{\text{PRF}}\right)(0:M-1)} \tag{11}$$

Taper vector $t$ is formed by the Kronecker product of the Doppler vector $d(w)$ and an identity vector of length $N$.

$$t = d(w) \bigotimes \text{ones}(1, N) \tag{12}$$

Prior to the covariance matrix estimation, space–time data $x$ in each range bin is tapered with the vector $t$.

$$x_c = x \left(\cdot\right) t \tag{13}$$

Tapering of the input data $x$ achieves a clutter ridge alignment for training range bins and weight vector can be computed by (10).

Performance evaluation of any linear processor can be carried out by analyzing SNR performance. In the case of STAP, signal to interference and noise ratio (SINR) loss and improvement factor (IF) are pertinent metrics. SINR loss is the SINR normalized to maximum achievable SINR. IF is a ratio of the output SINR to the input SINR. SINR loss and IF are given by (14) and (15) below, where $\xi_t$ is target SNR and $\xi_i$ is the input SINR.

$$\text{Loss}_{\text{SINR}} = \frac{\left(v^H R_c^{-1} v\right)}{\xi_t MN} \tag{14}$$

$$\text{IF}_{\text{SINR}} = \frac{\left(v^H R_c^{-1} v\right)}{MN\xi_i} \tag{15}$$

In this paper, we characterized the performance of the STAP algorithm in terms of target detection and parameters estimation. The primary task of the radar is to locate the position, estimate the velocity and bearing of a particular target. In order to estimate target parameters, the weight vector is computed for certain angles of interest and all Doppler bins. The angle vector covers the area illuminated by the main lobe of the radar antenna. The weight vector is applied to each range bin for all Doppler bins and angles of interest, resulting in a 3D array $S$. A flow graph illustrating the parameter estimation method is depicted in Figure 8.

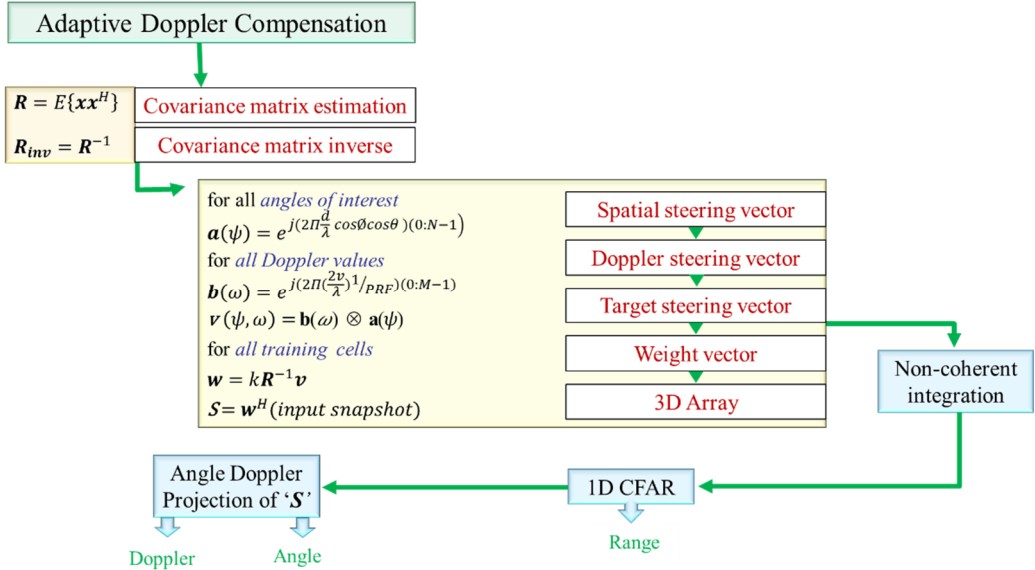

**Figure 8.** Target parameter estimation.

Non-coherent integration of *S* in angle and Doppler dimensions reduced it into a 1D range vector. The CFAR algorithm was applied to the 1D vector to detect the target range bin. Angle–Doppler projection of the detected target range bin in *S* was subsequently used for the angle of arrival and velocity estimation. In the angle–Doppler domain, the spatial dimension was along the rows while the temporal dimension was along with the columns. The peak value in the angle–Doppler spectrum indicates the target angle and velocity.

## 5. Results

### 5.1. Simulation Parameters

An X-band airborne phased array radar was modeled in Matlab. Radar parameters are given in Table 1. The radar pulse repetition frequency (PRF) and the platform velocity were adjusted to operate in the unambiguous regime. The radar antenna array comprised 8 Vivaldi antenna elements in the form of a ULA. The total number of pulses M in the CPI was 8. The radar platform height was set to 1000 m. Clutter was modeled using the Constant Gamma model with a gamma value of −15 dB Target was positioned at a distance of 3000 m with an azimuth angle of $0°$ and elevation angle of $−18.9°$. The target velocity was only 5 m/s. It is improbable to detect such a target with conventional signal processing, since it would be buried inside the clutter.

**Table 1.** Simulation parameters.

| Parameter | Value |
| --- | --- |
| Operating wavelength ($\lambda$) | 0.03 m |
| Pulse Repetition Frequency (PRF) | 10 kHz |
| Pulse Width (PW) | 0.2 μs |
| Sampling Rate ($F_s$) | 5 MHz |
| Number of antenna elements ($N$) | 8 |
| Number of pulses ($M$) | 8 |
| Number of range bins ($L$) | 500 |
| Crab angle $\gamma$ | $90°$ |
| Depression angle $\delta$ | $18.9°$ |

### 5.2. Range Doppler Map of Raw Data

Target was deliberately placed in the near range since STAP performance was most degraded in that region. The normalized Doppler frequency of target was 0.46, which placed the target at the edge of the main lobe clutter in the Doppler dimension. The range–Doppler map of unprocessed data is depicted in Figure 9. Extravagant Doppler variation starting from 1000 to 5000 m was evident that made the training samples non-homogeneous. Moreover, the strong clutter masked the target.

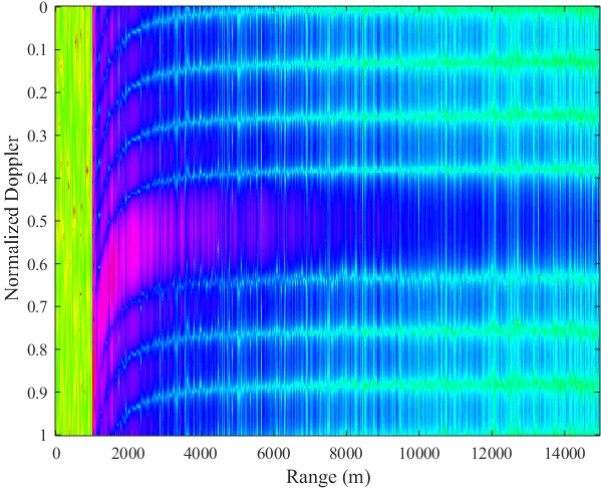

**Figure 9.** Range–Doppler map of raw data.

### 5.3. Estimation of Clutter Doppler Frequency

The Doppler frequency for each range bin was computed by employing the FFT operation and averaging over the 8 channels resulted in the net Doppler estimate. The theoretical value of Doppler for each range bin was computed by (5). Comparative analysis of theoretical and estimated Doppler frequency is given in Figure 10. The estimated Doppler frequency values closely match the theoretical ones. The Doppler frequency values have been given for the range bins starting from the altitude line.

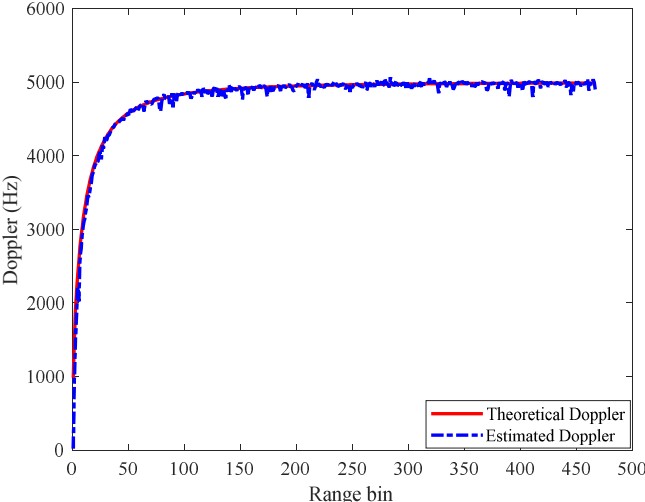

**Figure 10.** Theoretical and estimated Doppler frequency.

### 5.4. Angle Doppler Response after Adaptive Doppler Compensation

The raw data was adaptively Doppler compensated and the weight vector was computed as given by (10). The angle–Doppler response of the weight vectors after Doppler compensation is shown in Figure 11. Compared to Figure 6, it can be observed that the clutter null width was substantially decreased. A narrow clutter width improved the MDV of STAP. The wide clutter as depicted in Figure 6 attenuates slow-moving targets, which was substantially improved to detect very small moving targets for forward-looking radar geometry.

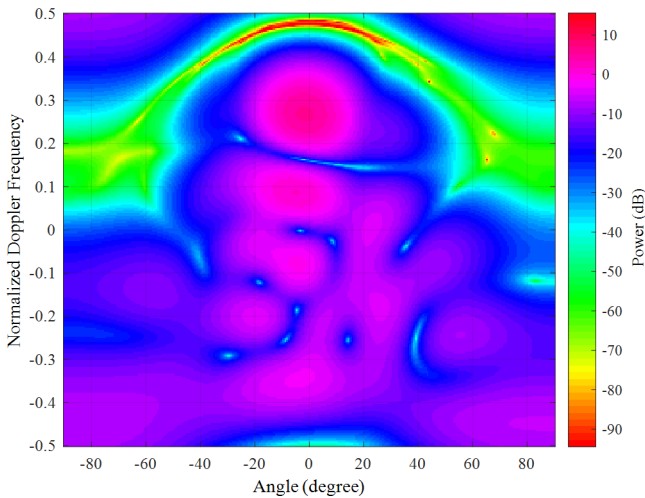

**Figure 11.** Weight vector response after adaptive Doppler compensation.

### 5.5. Target Parameter Estimation

The radar antenna beamwidth in the azimuth was $13°$ while it was $73°$ in the elevation direction. It was assumed that the target elevation angle is known. Target search in azimuth direction was undertaken from $-7$ to $+7°$. The actual target range bin index was 107. Since the dimension of space–time snapshot was $64 \times 1$, a total of 128 training range bins and 04 guard cells were utilized for covariance matrix estimation. A 3D array was obtained after applying the weight vector for all the Doppler frequencies and angles. The processed 3D data matrix is depicted in Figure 12, in the form of a volume plot. It may be appreciated that interference in all the range bins was efficiently suppressed. The two rectangular slices in the volume represent the range bin of the altitude line and the simulated target. In a real radar system, the altitude line return can be masked, given the platform height information is made available. In order to estimate the target parameters, 1D and 2D projection of the 3D array were utilized.

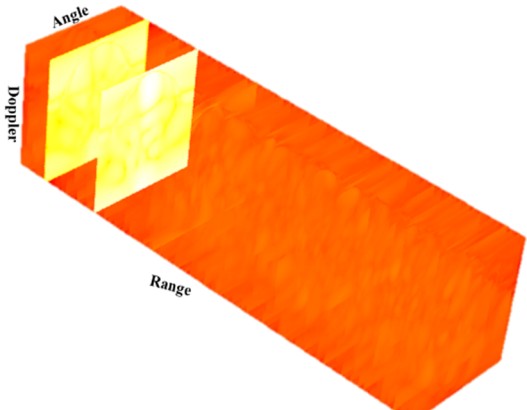

**Figure 12.** 3D processed data.

The 3D matrix was non-coherently integrated along Doppler and angle dimensions to obtain 1D range data. 1D range processed data is depicted in Figure 13. The estimated target range bin with adaptive Doppler compensation is 107, which is also its actual location. A peak visible at range bin 33 is the altitude line. STAP weights were not applied to the initial 41 range bin.

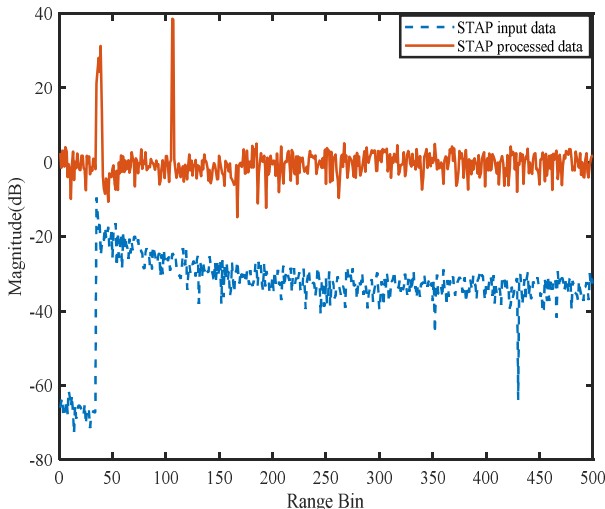

**Figure 13.** 1D range processed data.

It is highlighted that the angle–Doppler response of Figure 11 was obtained assuming that target Doppler and angle of arrival are known. In order to estimate the Doppler frequency and angle of arrival of the detected target, angle–Doppler projection of the 3D processed data is required. Figure 14 shows

the corresponding angle–Doppler projection of the 3D processed data at the detected range bin. It can be observed that clutter null width was narrow and the target amplitude was amplified. The estimated target azimuth angle was −0.7° while the actual value was 0°. It was found during multiple iterations that the estimated target angle varied from 0 to 2°. Accuracy in the angle estimate was predicated by the number of antenna elements. Since the number of antenna elements was only 8, low accuracy in angle estimation was expected.

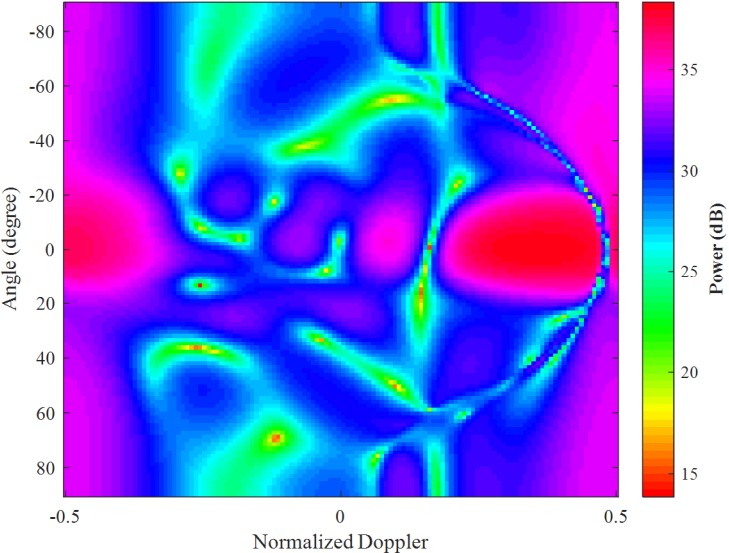

**Figure 14.** Angle–Doppler projection of 3D processed data.

The estimated target normalized Doppler was 0.3582 while the actual target Doppler was 0.3504. Range Doppler projection of the 3D processed data is depicted in Figure 15. A deep clutter null along the clutter Doppler frequency 0.46 was formed. The target can be noticed at range bin 107 and corresponding normalized Doppler. The estimated Doppler value also depended on the CPI length. Increasing the number of pulses may increase the Doppler resolution and subsequently the Doppler estimate. The accuracy of Doppler and angle estimates can also be improved by increasing the number of Doppler and spatial steering vectors. However, the real time processing requirements limit the maximum number of search vectors in a single CPI.

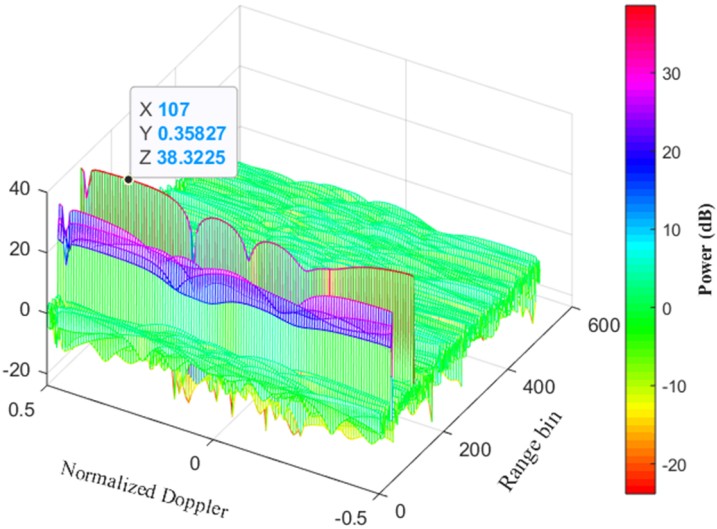

**Figure 15.** Range–Doppler projection of 3D processed data.

The range–Doppler projection may be required to implement 2D CFAR. It was found that using normalization constant $k = 1/v^H R_c^{-1} v$ decreased the number of false alarms, thereby improving the accuracy of estimated parameters.

It is pertinent to highlight that parameter estimation was also carried out for sample matrix inversion (SMI) without data compensation. Targets in near range bins and at skirts of the main lobe clutter in the Doppler dimension were attenuated by SMI without any data compensation.

### 5.6. STAP Metrics

The performance of the proposed adaptive Doppler compensation algorithm was also evaluated in terms of SINR loss. The covariance matrix was estimated for range bin 107 with a total of 128 training samples and 04 guard cells. The covariance matrix was separately computed for SMI without any data compensation, SMI with Doppler warping and SMI with adaptive Doppler compensation. SINR loss for the stated techniques is illustrated in Figure 16. SMI performance near clutter Doppler is severely degraded prior to any data compensation. Doppler warping and proposed adaptive Doppler compensation augment the SINR up to 25 dB. Performance of the proposed algorithm is almost equivalent to Doppler warping. However, radar parameters are not required to be known a priori for the adaptive Doppler compensation. In an actual scenario, Doppler warping performance may suffer due to the non-availability of radar parameters.

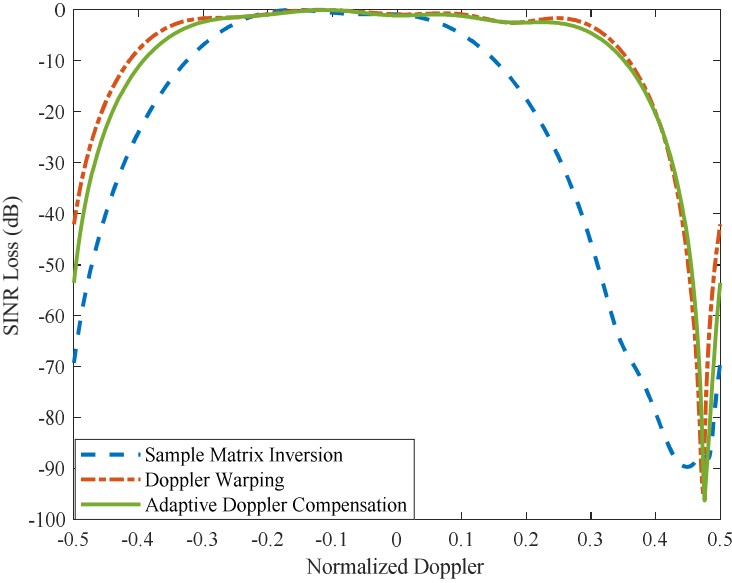

**Figure 16.** SINR loss.

### 5.7. Future Research Work

In this study, adaptive Doppler compensation was undertaken for SMI, which improved the degraded STAP performance. The results presented in this research assumed ULA, which limited angle estimation in the azimuth axis only. Furthermore, platform velocity was restricted so as to make clutter unambiguous in velocity.

It is proposed that current work may be expanded by incorporating a uniform planar array. It is expected that elevation degrees of freedom can be utilized to suppress range ambiguous clutter. CPI length may be increased and diagonally loaded SMI may be implemented. Increase in CPI length decreased correlation and hence diagonal loading might help ameliorate the decorrelation effects. Moreover, computation cost could be further curtailed by extending the parameter estimation method to RD-STAP algorithms.

## 6. Conclusions

In this study, an airborne phased array radar was modeled to generate radar synthetic data. The clutter covariance matrix was estimated from the synthetic data. The clutter Doppler dependence on range degraded the STAP performance in near ranges. The adaptive Doppler compensation was proposed to ameliorate the degraded performance. The validity of the proposed method was corroborated through the estimation of target parameters. Moreover, the performance of the proposed algorithm successfully verified its efficacy in terms of SINR loss. The proposed method yielded average performance improvement of 25 dB relative to traditional STAP SMI.

**Author Contributions:** Conceptualization, M.B.K. and A.H.; methodology, M.B.K. and A.H.; software, A.H.; validation, A.H., U.A. and C.B.A.; data curation, C.B.A.; writing—original draft preparation, A.H.; writing—review and editing, M.B.K. and A.H., supervision, X.Y.; project administration, X.Y.; funding acquisition, X.Y.; All authors have read and agreed to the published version of the manuscript.

**Funding:** National natural science Foundation of China 61301175.

**Conflicts of Interest:** The authors declare no conflict of interest.

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
