# Peer review of "Adaptive Doppler Compensation for Mitigating Range Dependence in Forward-Looking Airborne Radar"

_electronics, doi:10.3390/electronics9111896_

Round 1

Reviewer 1 Report

This paper presents an adaptive Doppler compensation scheme. It would be better to improve the quality of the paper by considering the following concerns in the revision.

  1. The first letters of the words abbreviated do not have to be capitalized.
  2. When referring to equations, simply use “(x)”, not “Equation (x)”.
  3. 2 is not a vector type and of low quality.
  4. The references should include recent works, and the proposed scheme should be compared with those.
  5. Table 1 shows only one set of parameters simulated. Is it possible to examine for another set to generalize the results?
  6. For the algorithm in Fig. 5, it would be beneficial to add an example with practical values.
  7. Throughout the paper, English needs to be improved.

Author Response

Reply to reviewer’s comments

We are thankful to the editor for giving us the opportunity to revise and resubmit the manuscript. We are also grateful to the worthy reviewer for his valuable comments which helped to improve the quality of the manuscript. We have introduced our responses to the reviewers’ comments below. The text in black is the reviewer’s comment and in blue italic is the author’s response. Line numbers are also changed due to the addition and deletion of text, figures and tables.

Following documents are attached along with this response letter

  • Revise highlighted manuscript (Using MS word Track changes function)

Description:

In this document, we did track changes using MS word function in order to address the reviewer’s suggestions. We add and delete sentences, modify figures tables and update references. Proofreading by eliminating grammatical mistakes and typos errors.

  • Corrected manuscript

Description:

In this document, we accept all the changes and stop tracking. We correct the spaces and typo errors in this final corrected manuscript. This is the final version for submission because all new changes are also in this version.

Reply to reviewer’s comments

1- The first letters of the words abbreviated do not have to be capitalized.

Reply:

We are grateful to worthy reviewer, the capitalized words have been corrected.

2-            When referring to equations, simply use “(x)”, not “Equation (x)”.

Reply:

We are grateful to worthy reviewer, the word ‘Equation ’ inside the text has been deleted.

3-  2 is not a vector type and of low quality.

Reply:

I could not exactly isolate the referred suggestion in the text.

4- The references should include recent works, and the proposed scheme should be compared with those.

Reply:

The references have been updated to the latest publications. Furthermore, the performance of  proposed adaptive Doppler compensation has been ccompared with the Doppler warping method and Sample Matrix Inversion algorithm.

5- Table 1 shows only one set of parameters simulated. Is it possible to examine for another set to generalize the results?

Reply:

The important parameters highlighted in table include PRF and crab angle. The proposed algorithm mitigates range dependance of  clutter Doppler for a forward looking radar. We can change the radar geometry (vary crab angle between 0 to 90) to any non side looking array. Furthermore PRF of the radar can also be varied. We varied these parameters to test the efficacy of the algorithm and it was found that the proposed algorithm performs well for any non-sidelooking geometry. 

6- For the algorithm in Fig. 5, it would be beneficial to add an example with practical values.

Reply:

The algorithm depicted in Figure 5 (figure 8 in the new manuscript) presents target parameter estimation method. In the paper, the target range(3 km), Doppler(0.46) and azimuth angle (0 deg) have been estimated by the algorithm presented in the referred figure. The simulation parameters have been kept same for the Doppler compensation algorithm and parameter estimation.   

  1. Throughout the paper, English needs to be improved

Reply:

The abstract of the paper and introduction has been enhanced and rephrased to improve the logical structure of the paper. Moreover, some of the sentences have also been rephrased.

Reviewer 2 Report

see attachment

Author Response

Reply to reviewer’s comments

We are thankful to the editor for giving us the opportunity to revise and resubmit the manuscript. We are also grateful to the worthy reviewer for his valuable comments which helped to improve the quality of the manuscript. We have introduced our responses to the reviewers’ comments below. The text in black is the reviewer’s comment and in blue italic is the author’s response. Line numbers are also changed due to the addition and deletion of text, figures and tables.

Following documents are attached along with this response letter

  • Revise highlighted manuscript (Using MS word Track changes function)

Description:

In this document, we did track changes using MS word function in order to address the reviewer’s suggestions. We add and delete sentences, modify figures tables and update references. Proofreading by eliminating grammatical mistakes and typos errors.

  • Corrected manuscript

Description:

In this document, we accept all the changes and stop tracking. We correct the spaces and typo errors in this final corrected manuscript. This is the final version for submission because all new changes are also in this version.

Reviewer Comments:

1- In my advice, this work could be of some interest to the reader, and I have found it quite well structured. Nevertheless, I recommend a major revision of this work. In my opinion, this paper is excessively concise and addressed to a public of specialists, so it is the understanding of the proposed approach become too difficult; moreover sometimes also English language usage appears too “heavy”. Result analysis too in my opinion suffer of the same problem, with many details but a difficulty to appreciate the improvements of the proposed approach. On the other hand, abstract must be rewritten, for the same reasons (especially the first part), while introduction could be expanded to better explain the problems addressed by the proposed

approach.

My general advice is to review the entire paper, expanding details, better presenting results and focusing on a public of less specialist scientists.

Reply: The abstract of the paper has been rewritten and exteneded so as to make the content easy to understand for the audience. Likewise the introduction of the paper has been elaborated and some of the latest references have been included. It may be noted that a complete STAP framework encompassing clutter suppression, target detection and parameter estimation have been presented in this paper. Although, the problem of Doppler compensation has already been addressed in the literature but the proposed adaptive Doppler compensation method circumvents the need of radar parameters in real time. Performance comparison of the proposed doppler compensation method with the well established algorithms has been illustrated in iterms of SINR loss (Figure 16). Moreover, as per the knowledge of author,the proposed  parameter estimation method is a novel idea that includes non-coherent integration followed by 1D CFAR to detect and estimate range and peak isolation in the angle-Doppler response.

2- Other point that in my opinion has to be better addressed is the simulated scenery. The discussion is on forward-looking airborne radars: do you retain interesting a small discussion on other type of radars? In this respect, which are the limits of the proposed approach?

Reply:

Thanks for the highlighting this point. Referring to figure 1 in the latest manuscript, the problem of range dependence exists in almost all geometries except a side looking one. Moreover, the Doppler dependence on range is highest in case of a forward looking radar. The proposed adaptive Doppler compensation is applicable for all the non side looking geometries and results hold equally true. The only difference between any two non sidelooking geometries is only the rate of change of frequency with the range.

As far as the limitations are concerned, the proposed algorithm assumes clutter to be unambiguous in range and doppler. Furthermore, a uniform linear array is considered which limits angle estimation in 1D only.   

3- The simulated scenery is described, with many details, yet I find difficult to compare it with a real world scene. Moreover, could you try to discuss the impact of a real world scene respect the proposed simulation?

Reply:

The proposed simulation mimics an X-band radar mounted as a nose radar in an airborne platform. The selection of PRF and platform velocity have been done so as to keep the clutter unambiguous in velocity. The airborne radar is illuminting a target at a range of 3 km with an azimuth angle of 0 degree and elevation angle of -18.9 degree. The target velocity is only 5 m/s. With conventional radar signal processing, such a target would be buried inside a clutter. However, due to the proposed adaptive Doppler compensation scheme, the minimum detectable velocity of STAP algorithm is improved and target is detected even with such a small velocity.

4- As minor comments, I have some doubts on the correctness of figure 1 (especially the v-axis) and I suggest a serious revision of figure 5.

Reply:

The figure 1(figure 3 in revised manuscript) was rechecked for correctness. Figure 5(figure 8 in revised manuscript) was redone to improve clarity.

.

Round 2

Reviewer 1 Report

The reviewer is sorry for the previous comment including a typo. It was on Fig. 2, which is now Fig. 4. The figure is still of low quality, and it would be better if it can be converted to a vector graphic, not a bitmap one. The rest of the concerns have been addressed well. Thank you.

Author Response

We are very grateful to the reviewer for appreciating our work and give valuable suggestions. We attached our response below:

Reviewer 2 Report

Dear Authors, i have appreaciated the excellent improvements in introduction, abstract and references and also the new fig. 8. In my opinion you should make an effort also to further improve clarity of the technical sections 2 - 6. Moreover i retain that you should report in the paper your answears to my previous questions. Finally, Fig. 3 in my advice is still unclear: what represents the dotted line and dots?

Author Response

(The authors gave the same response as above.)
